Identification of potential obese-specific biomarkers and pathways associated with abdominal subcutaneous fat deposition in pig using a comprehensive bioinformatics strategy

Yang Yongli 1
Wang Xiaoyi 1
Li Mingli 1
http://orcid.org/0000-0003-0380-5798 Wang Shuyan 1
Wang Huiyu 1 2
Chen Qiang 1 chq@sjtu.edu.cn
Lu Shaoxiong 1 shxlu_ynau@163.com
1 Faculty of Animal Science and Technology, Yunnan Agricuture University , Kunming , China
2 Faculty of Animal Science, Xichang University , Xichang , China
Uversky Vladimir
Electronic publication date: 2024 May 31
Publication date: 2024
Volume: 12
Electronic Location ID: e17486
Received 2023 Dec 8; Accepted 2024 May 8
Copyright: © 2024 Yang et al.
Copyright year: 2024
Copyright holder: Yang et al.
License: This is an open access article distributed under the terms of the Creative Commons Attribution License, which permits unrestricted use, distribution, reproduction and adaptation in any medium and for any purpose provided that it is properly attributed. For attribution, the original author(s), title, publication source (PeerJ) and either DOI or URL of the article must be cited.
License URL: https://creativecommons.org/licenses/by/4.0/

Keywords: Abdominal subcutaneous fat deposition, Pig, Hub gene, Biomarkers, Key pathway

Funding: National Key Research and Development Program of China 2022YFD1601902 Yunnan Swine Industry Technology System Program 2020KJTX0016 Yunnan Province Important National Science & Technology Specific Projects 202104BI090022, YNWR-CYJS-2018-056, 202102AE090039 This research was supported by the National Key Research and Development Program of China (2022YFD1601902), the Yunnan Swine Industry Technology System Program (2020KJTX0016), and the Yunnan Province Important National Science & Technology Specific Projects (202104BI090022, YNWR-CYJS-2018-056, 202102AE090039). The funders had no role in study design, data collection and analysis, decision to publish, or preparation of the manuscript.

==============================
Abdominal subcutaneous fat deposition (ASFD) is not only related to meat quality in the pig industry but also to human health in medicine. It is of great value to elucidate the potential molecular mechanisms of ASFD. The present study aims to identify obese-specific biomarkers and key pathways correlated with ASFD in pigs. The ASF-related mRNA expression dataset GSE136754 was retrieved from the Gene Expression Omnibus (GEO) database and systematically analyzed using a comprehensive bioinformatics method. A total of 565 differentially expressed genes (DEGs) were identified between three obese and three lean pigs, and these DEGs were mainly involved in the p53 signaling pathway, MAPK signaling pathway and fatty acid metabolism. A protein-protein interaction (PPI) network, consisting of 540 nodes and 1,065 edges, was constructed, and the top ten genes with the highest degree scores—ABL1, HDAC1, CDC42, HDAC2, MRPS5, MRPS10, MDM2, JUP, RPL7L1 and UQCRFS1—were identified as hub genes in the whole PPI network. Especially HDAC1, MDM2, MRPS10 and RPL7L1 were identified as potential robust obese-specific biomarkers due to their significant differences in single gene expression levels and high ROC area; this was further verified by quantitative real-time PCR (qRT-PCR) on abdominal subcutaneous fat samples from obese-type (Saba) and lean-type (Large White) pigs. Additionally, a mRNA-miRNA-lncRNA ceRNA network consisting of four potential biomarkers, 15 miRNAs and 51 lncRNAs was established, and two targeted lncRNAs with more connections, XIST and NEAT1, were identified as potentially important regulatory factors. The findings of this study may provide novel insights into the molecular mechanism involved in ASFD.

Introduction

Fat deposition is an important economic trait in pig production. The amount of fat deposition not only affects growth rate and feed conversion efficiency but also influences lean meat rate, pork quality, and reproductive performance (Zhang et al., 2019). Moreover, swine and human share a high degree of homology in genome sequence and chromosome structure, as well as higher similarities in physiological characteristics. Pigs are often used as biomedical models to study obesity-related diseases such as type 2 diabetes mellitus, cardiovascular disease, and certain types of cancer (Robich et al., 2010). Thus, it is highly valuable to reveal the molecular mechanism involved in fat deposition.

Fat deposition is a dynamic process that involves fat synthesis, decomposition, and transportation (Samra, 2000). Some published studies showed that fat deposition rates varied among different pig breeds (Cui et al., 2019; Wang et al., 2020). Subcutaneous fat (SCF), which mainly includes back subcutaneous fat (BSF) and abdominal subcutaneous fat (ASF), is not only a primary energy storage organ but also a main endocrine organ. SCF plays important regulatory roles in many biological and physiological processes such as body temperature regulation, energy balance, insulin sensitivity, inflammatory reaction, and cardiovascular reaction (Sauerwein et al., 2014). Further, studies have shown that SCF content influenced carcass characteristics and was closely associated with pork quality (Verbeke et al., 1999). In particular, BSF content has been used as an evaluation index for predicting the lean meat percentage in pigs (Sauerwein et al., 2014). At present, most studies mainly focused on the molecular mechanisms of BSF deposition (BSFD) (Hou et al., 2016; Zhang et al., 2007), and some important genes and key signaling pathways correlated with BSFD were identified in pigs (Sodhi et al., 2014; Zambonelli et al., 2016; Zhang et al., 2022). However, little attention was paid to the molecular mechanisms of ASF deposition (ASFD). Some published studies showed that ASF tissue (ASFT) not only stored energy, but more importantly, acted as an endocrine organ to secrete large amounts of inflammatory mediators and more favorable adipokines in human, and might be associated with the pathophysiology of obesity complications, such as insulin resistance (Abate et al., 1996; Chait & den Hartigh, 2020; Ibrahim, 2010). For example, adipocytokines including IL-6, leptin and visceral adiponectin were highly expressed in ASFT than in visceral fat, which indicated that ASFT might have an important role in the regulation of systemic inflammation (Cheng et al., 2008). Despite some advances in ASFT (Pawar et al., 2015; Yang et al., 2010), our understanding of the role of ASFD in biological processes is still limited, and the hub genes and key pathways involved in ASFD still need to be explored.

In the present study, mRNA expression profiles related to ASFT (GSE136754) from obese and lean pigs were retrieved from the Gene Expression Omnibus (GEO) database and systematically analyzed to identify hub genes and key pathways using a comprehensive bioinformatics method, including differentially expressed gene analysis (DEGA), functional enrichment analysis, gene set enrichment analysis (GSEA) and protein-protein interaction (PPI) construction. Subsequently, potential obese-specific biomarkers were identified among these hub genes based on single gene expression levels and ROC curve values, and their functions were investigated using the GSEA method. Furthermore, the regulatory relationships of mRNA (potential obese-specific biomarkers)-miRNA-lncRNA were elucidated by constructing a ceRNA interaction network, and the transcription factors (TFs) were predicted. Furthermore, the potential obese-specific biomarkers were further validated by qRT-PCR in obese- and lean-type breeds, Saba and Large White pigs. This study will provide novel insights into the functional roles of ASFT in biological processes. Portions of this text were previously published as part of a preprint (Yang et al., 2022).

Materials and Methods

Data collection and preprocessing

The gene expression dataset GSE136754 was retrieved from the public GEO database (GPL11429 Illumina HiSeq 2000 (Sus scrofa) (https://www.ncbi.nlm.nih.gov/geo/) and generated using mature adipocytes (MAs) from crossbred F2 pigs (Duroc × Göttingen minipig) (Jacobsen et al., 2019). A total six samples (three lean pigs and three obese pigs) were used for our study. The original data were transformed into the transcription per million (TPM) reads for the following analyses (Table S1), the equation for calculation is as follows:

TPMi=FPKMi/εFPKMj.106.

The overview of the workflow is shown in Fig. 1.

Figure 1 Flow chart of the bioinformatics analysis in the present study.

Identification of differentially expression genes

To identify key genes involved in ASFD, a DEGA was implemented between ASFTs from the obese and lean pigs by the Limma package in R software (version 4.1.2) (Ritchie et al., 2015). Genes with a |log2FC (fold change) | >1.5 and a P-value < 0.05 were considered as DEGs. The expression level and distribution of differentially expressed genes (DEGs) were visualized using a heatmap and a volcano map based on the ggplot2 package (versions 3.3.6), respectively.

Functional enrichment analysis of DEGs

To investigate the biological functions of DEGs, functional enrichment analyses including GO and KEGG pathway analyses were performed using the clusterProfiler package (version 4.4.4) in the Bioconductor project (version 3.15) (Kanehisa & Goto, 2000). The GO term includes three categories, cellular component (CC), biological process (BP), and molecular function (MF). Functional terms with a P-value < 0.05 were considered to significantly changed, and the top ten functional terms were visualized using the ggplot2 package.

GSEA

To further investigate the function of identified genes, a GSEA was performed to identify the key pathways with biological processes GO and KEGG annotation pathway sets using the clusterProfiler package and GSEABase (version 1.58.0) R software, and the results were visualized using the enrichplot package (version 1.16.1) (Yu et al., 2012). Briefly, the expression levels of all genes using logFC sequencing were used in GSEA and then the enrichment scores were calculated according to the ranked-ordered gene list. The significance scores were computed based on the 10,000 nonparametric permutations tests of gene sets. Gene sets with normalized enrichment score (NES) >1 and a nominal P-value < 0.05 were regarded as significant enrichment gene sets.

PPI network construction and hub gene identification

To elucidate the interactive relationships among DEGs encoding proteins, a PPI network was constructed using the online STRING database (https://cn.string-db.org/) (Szklarczyk et al., 2015), and visualized by the Cytoscape software (version 3.9.1) (Shannon et al., 2003). Subsequently, the highly correlated module was extracted from the whole PPI network using the Molecular Complex Detection (MCODE) algorithm in the Cytoscape MCODE plugin (version 2.0.2) (Bader & Hogue, 2003). To ensure the validity of the screening results, the threshold parameters for the module genes were set as follows: degree cutoff = 2, node score cutoff = 0.2, k-score = 2, and max. Depth = 100. The module that obtained the highest score was identified as the most critical module. Hub genes were identified using a degree centrality method in the Cytoscape plugin cytoNCA (version 2.1.6) (Tang et al., 2015), and the top ten genes with the highest score were selected as hub genes. The interaction network of hub genes along with GO and KEGG terms was visualized by the ClueGo (version 2.5.9) and CluePedia plugins (version 1.5.9) (Bindea, Galon & Mlecnik, 2013; Bindea et al., 2009).

Identification of potential obese-specific biomarker

To identify the potential obese-specific biomarker, the predictive performance of hub genes was evaluated using an unpaired t-test to assess the gene expression. A gene with a P-value < 0.05 was considered as a potential biomarker with significant change. The efficacy of the predictive performance of potential biomarkers was assessed by establishing receiver operating characteristic (ROC) curves and calculating area under the ROC curve (AUC) values using GraphPad Prism (version 8.0.1). A gene with an AUC > 0.7 was considered to be a potential obese-specific biomarker, typically regarded as having medium accuracy in the range of 0.7 to 0.9 and high accuracy above 0.9.

Construction of ceRNA network

To explore the potential regulatory relationships among obese-specific biomarkers, targeted miRNAs and lncRNA, a mRNA-miRNA-lncRNA ceRNA network was constructed. The potential targeted miRNAs and lncRNAs of biomarkers were predicted through the ENCORI database (https://rna.sysu.edu.cn/encori/index.php). First, the mRNA-miRNA relationship pairs were extracted based on at least two of the databases, miRanda (John et al., 2004), TargetScan (Agarwal et al., 2015), RNA22 (Loher & Rigoutsos, 2012), and miRmap (Vejnar & Zdobnov, 2012), that is, the intersection of predicted results from at least two out of these four software was identified as final results. Then, the interacting lncRNAs were predicted according to these predicted miRNAs. Finally, the ceRNA network was constructed based on the miRNAs that intersect between the potential lncRNA-miRNA pairs and miRNA-mRNA pairs. A ceRNA network was visualized using the Cytoscape software.

Gene regulatory network analysis of obese-specific biomarker

The TFs of obese-specific biomarkers were predicted through the online database of UCSC (https://genome.ucsc.edu/) and JASPAR 2022. The minimum score = 800, which is a strict criterion (default criteria: 200), was set as the screening threshold. This means that only the TFs with scores above 800 were shown. The regulatory relationships between TFs and potential biomarkers were visualized through the Cytoscape software (version 3.9.1).

Animals and tissue collection

Three Chinese indigenous obese-type Saba pigs and three Western lean-type Large White pigs were used to validate the expression patterns of selected potential obese-specific biomarkers, which had no related kinship with each other. These pigs were obtained from the national-level Saba pig conservation farm (Chuxiong City, Yunnan Province, China). They were raised in two growth stages, 25–60 and 60–100 kg liveweight. All the pigs were fed with the same diet at same growth stage, and the main nutrients of the diets were shown in Table 1. The pigs had access to food and water ad libitum, and were maintained under same conditions until slaughter (~100 kg). The pigs were fasted overnight before the slaughter with free access to water. The pigs were slaughtered via electroshock followed by exsanguination. The tissues including ASF that is the inner layer of subcutaneous fat on the abdominal fat, muscles and organs were collected, snap frozen in liquid nitrogen and maintained at −80 °C until subsequent analysis. And the ASF tissues were used for this study. The animal experiment was conducted in compliance with the principles stated in the guide for the regulations for the Administration of Affairs Concerning Experimental Animals (Ministry of Science and Technology, China, revised in June 2004). The experimental protocol was approved by the Animal Ethics Committee of Yunnan Agricultural University (approval ID: 202310003).

Table 1 Main nutrient ingredients of the diets for different growth stages of obese-type Saba pigs and Western lean-type Large White (25–60 and 60–100 kg liveweight).

All pigs had the same feed ad libitum at same growth stage.

Growth stage	25–60 kg	60–100 kg	
Crude protein, %	15.78	16.58	
Digestible energy, MJ/kg	13.23	13.35	
Crude fiber, %	3.50	3.50	
Calcium, %	0.65	0.52	
Total phosphorus, %	0.56	0.48	
Lysine, %	0.75	0.63	
Methionine, %	0.45	0.27	
Threonine, %	0.95	0.60	
Valine, %	0.95	0.65	

RNA extraction and qRT-PCR

Total RNA was extracted from ASF tissue samples using the RNA sample total Extraction Kit (Tiangen, Beijing, China). Reverse transcription was performed using PrimeScript™ RT reagent Kit with gDNA Eraser (Takara, Dalian, China) according to the manufacturer’s instructions. qPCR assay was performed using TB Green® Premix Ex Taq™ II (Tli RNaseH Plus) (Takara, Dalian, China) on a qPCR system (Mx3000P, Agilent Technologies, Santa Clara, CA). The gene-specific qPCR primers are listed in Table S2. Each experiment was performed in triplicates, and relative expression of mRNA was calculated though the 2−ΔΔCt method, GAPDH was used as the internal control for normalization, the equation for calculation is as follows: ΔCt = target gene Ct—GADPH Ct, ΔΔCt = test gene ΔCt - control ΔCt, and gene expression = 2(ΔΔCt). The relative mRNA expression levels of obese- and lean-type pigs were compared by an unpaired t-test using SAS software (version 9.2), and P < 0.05 was considered significant while P < 0.01 was statistically significant.

Results

Identification of DEGs

The totals of 565 DEGs were identified (P < 0.05) by DEGA between ASFTs from obese and lean pigs, including 412 up- and 153 down-regulated genes in the MAs of ASFT from obese pigs compared with lean pigs (Fig. 2A, Table S3). The expression levels of these DEGs were presented using a heatmap in Fig. 2B.

Figure 2 Identification and functional analysis of DEGs.

(A) Volcano plot of the expression level of differentially expressed genes in obese and lean pigs GSE136754. Red and dots represent high and low expression of genes, respectively. The dots with blackcircle were hub genes. (B) Heatmap of the expression level of differential expressed genes between obese and lean pigs from GSE136754; the abscissa indicates the sample names, and the ordinate shows the gene names. (C–E) The top 10 terms significantly enriched in the three GO categories (C) biological process (D) cellular component and (E) molecular function. (F) The top 10 terms significantly enriched in KEGG pathway. (G) GSEA analysis based on logFC sorting as a list of genes, with the GO–MF as the reference gene set. (H) GSEA analysis based on logFC sorting as a list of genes, with the KEGG pathway as the reference gene set.

Functional analysis of DEGs

GO analysis showed that a total of 131 BPs, 34 CCs, and 36 MFs were significantly enriched (P < 0.05) (Table S4). The top 10 GO terms with the smallest P-values were shown in Figs. 2C–2E. Among the enriched BPs, DEGs were significantly enriched in the integrin-mediated signaling pathway (GO:0007229 and P = 0.0001), maintenance of protein location (GO:0045185 and P = 0.0021), regulation of binding (GO:0051098 and P = 0.0024), maintenance of location (GO:0051235 and P = 0.0026), ubiquitin-dependent protein catabolic process (GO:0006511 and P = 0.0031), and fatty acid beta-oxidation (GO:0006635 and P = 0.0032) (Fig. 2C). The CCs were mainly involved extracellular organelle (GO:0043230 and P = 8.81E-06), extracellular membrane-bounded organelle (GO:0065010 and P = 8.81E-06), extracellular vesicle (GO:1903561 and P = 8.81E-06), extracellular exosome (GO:0070062 and P = 4.87E-05), and ubiquitin ligase complex (GO:0000151 and P = 0.0023) (Fig. 2D). The MFs were mainly involved protease binding (GO:0002020 and P = 0.0003), signal sequence binding (GO:0005048 and P = 0.0010), actin binding (GO:0003779 and P = 0.0017), receptor tyrosine kinase binding (GO:0030971 and P = 0.0017) and enzyme binding (GO:0019899 and P = 0.0021) (Fig. 2E). KEGG enrichment analysis showed that a total of 41 KEGG pathways were significantly enriched (P < 0.05) (Table S5), and these DEGs were mainly implicated in p53 signaling pathway (ssc04115 and P = 0.0005), chronic myeloid leukemia (ssc05220 and P = 0.0006), proteoglycans in cancer (ssc05205 and P = 0.0007), fatty acid metabolism (ssc01212 and P = 0.0139), and MAPK signaling pathway (ssc04010 and P = 0.0142). The top 10 KEGG pathways with the smallest P-values were shown in Fig. 2F.

GSEA of DEGs

GO terms results showed that only MFs was enriched by the lowly expressed genes, and they were predominantly associated with cytokine activity (GO:0005125 and P = 7.60E-05), signaling receptor regulator activity (GO:0030545 and P = 0.0004), signaling receptor activator activity (GO:0030546 and P = 0.0005), and receptor ligand activity (GO:0048018 and P = 0.0005) (Fig. 2G, Table S6). No GO terms were enriched by the highly expressed genes. KEGG analysis showed that 29 pathways were significantly enriched by the highly expressed genes, while only two KEGG terms enriched in the lowly expressed genes (Table S6). The top five KEGG pathways enriched by the highly expressed genes were valine, leucine, and isoleucine degradation (ssc00280 and P = 0.0004), propanoate metabolism (ssc00640 and P = 0.0006), retinol metabolism (ssc00830 and P = 0.0053), one carbon pool by folate (ssc00670 and P = 0.0068), and glycosaminoglycan biosynthesis-chondroitin sulfate/dermatan sulfate (ssc00532 and P = 0.0121), respectively (Fig. 2H).

PPI network construction and hub genes identification

Among the 565 identified DEGs, there were 540 genes with 1,065 gene-gene interactive pairs. A PPI network consisting of 540 nodes and 1,065 edges was established (Fig. 3A). Highly correlated module analysis showed that 12 modules were identified in the entire PPI network, and the module with the highest score (score = 10.909) including 12 nodes and 120 edges (Fig. 3B). The 12 genes in the module were involved in the ribosome (ssc03010 and P = 1.25E-11), coronavirus disease-COVID-19 (ssc05171 and P = 1.951-06), and protein export (ssc05171 and P = 0.0231).

Figure 3 Biological analysis based on 565 DEGs.

(A) A PPI network, including 540 nodes and 1,065 edges. (B) Highly correlated module with the highest score in the whole PPI network (score = 10.909), including 12 nodes and 120 edges. (C) Significantly functional enrichment pathway of 10 hub genes.

In the whole PPI network, the top 10 genes with the higher degree score-ABL1, HDAC1, CDC42, HDAC2, MRPS5, MRPS10, MDM2, JUP, RPL7L1 and UQCRFS1 were identified as the hub genes (Table 2). Except for MRPS10 and RPL7L1, the other eight hub genes were up-regulated.

Table 2 The top 10 hub genes with the highest ranking identified by a degree centrality method in the Cytoscape plugin cytoNCA.

Num	Gene symbol	Entrez ID	logFC	P-value	Degree	Status	
1	ABL1	100,524,544	2.586311	0.01251	56	UP	
2	HDAC1	100,622,482	2.390058	0.000219	52	UP	
3	CDC42	780,428	5.900558	0.017737	52	UP	
4	HDAC2	100,156,170	3.085433	2.44E-05	50	UP	
5	MRPS5	100,521,262	2.402256	0.033097	46	UP	
6	MRPS10	100,155,549	−2.0789	0.000803	46	DOWN	
7	MDM2	100,125,959	1.684479	0.03648	46	UP	
8	JUP	397,592	1.695537	0.01367	44	UP	
9	RPL7L1	100,141,406	−2.10594	0.01976	40	DOWN	
10	UQCRFS1	NA	2.92074	0.00303	40	UP	

Biological functional of hub genes

Among the 10 hub genes, there were five hub genes, ABL1, CDC42, HDAC1, HDAC2 and JUP were significantly associated with some BPs including positive regulation of cell-matrix adhesion (GO:0001954 and P = 1.16E-04) and modification of synaptic structure (GO:0099563 and P = 1.22E-05). Two hub genes, HDAC1 and HDAC2 were significantly associated with some MFs including NAD-dependent histone deacetylase activity (GO:0034979 and P = 3.01E-05), histone deacetylase activity (GO:0031078 and P = 9.99E-06). Four hub genes, HDAC1, HDAC2, MRPS10 and MRPS5 were significantly correlated with some CCs including organellar small ribosomal subunit (GO:0000314 and P = 4.21E-05) and nuclear transcriptional repressor complex (GO:0090568 and P = 5.60E-05). KEGG pathway analysis revealed that four hub genes- ABL1, HDAC1, HDAC2 and MDM2 - were primarily associated with chronic myeloid leukemia (KEGG:05220 and P = 3.27E-08) and notch signaling pathway (KEGG:04330 and P = 2.38E-04). The interactive relationships between hub genes and GO/KEGG terms were shown in Fig. 3C and Table S7.

Identification of potential obese-specific biomarker

The single gene expression analysis in hub genes showed that HDAC1 and MDM2 were significantly up-regulated, but MRPS10 and RPL7L1 were significantly down-regulated in obese pigs. HDAC1, MDM2, MRPS10 and RPL7L1 were identified as vital genes among the ten hub genes (Figs. 4A–4J). And the AUCs of HDAC1, MDM2, MRPS10 and RPL7L1 were 0.88, 0.96, 0.96 and 1, respectively (AUC > 0.7), which showed these genes as obese-specific biomarkers had a higher effectiveness (Figs. 4A–4J). In terms of gene expression level, HDAC1 had a significant positive correlation with MDM2 and significant negative correlation with MRPS10 (Fig. 5A). GSEA results demonstrated consistency with DEGs functional analysis in pathways such as propanoate metabolism, ribosome, and C-type lectin receptor signaling pathway. Meanwhile, some pathways related with inflammation and immunity process, including butanoate metabolism, retinol metabolism, T cell receptor signaling pathway and oxidative phosphorylation were identified (Table S8). The top five significant pathways were shown in Fig. 6.

Figure 4 The evaluation of hub genes.

(A–J) Front: the expression level of the hub genes in the data set GSE136754. Behind: the ROC curve of the hub genes in the data set GSE136754. *P < 0.05, **P < 0.01, ns, no significant difference.

Figure 5 Analysis of obese-specific biomarkers.

(A) Correlations between potential obese-specific biomarkers. (B) The mRNA-miRNA-lncRNA ceRNA interaction network, including four potential biomarkers, 15 miRNAs and 51 lncRNAs. Biomarkers are presented in red triangle (upregulated genes) and dark green arrow (downregulated genes), whereas targeted miRNAs are shown in orange square circles and targeted lncRNAs are shown in yellow circles. (C) A comprehensive transcriptional regulatory network of potential biomarkers, including four potential biomarkers and 29 TFs. Biomarkers are presented in red triangle (upregulated genes) and dark green arrow (downregulated genes), whereas predicted TFs are shown in yellow circles. (D) Validation of mRNA expression levels of potential obese-specific biomarkers by qRT-PCR. The symbol * means significant difference, *P < 0.05, **P < 0.01.

Figure 6 GSEA functional analysis of potential obese-specific biomarkers.

(A) HDAC1. (B) MDM2. (C) MRPS10. (D) RPL7L1.

Construction of ceRNA network

Totals of 15 mRNA-miRNA and 113 miRNA-lncRNA relationship pairs were predicted. A ceRNA network including four mRNAs, 15 miRNAs and 51 lncRNAs was established (Fig. 5B). Among the lncRNAs, NEAT1 (13 miRNAs) and XIST (nine miRNAs) exhibited more connections with miRNAs.

Gene regulatory network analysis of obese-specific biomarkers

A total of 29 TFs targeting obese-specific biomarkers were predicted, and a comprehensive transcriptional regulatory network of hub genes was established (Fig. 5C). Among them, Two TFs—ZNF460 (four biomarkers) and ZNF384 (four biomarkers), had the most links with obese-specific biomarkers.

Validation of the biomarkers via qRT-PCR

The expression patterns of the four potential obese-specific biomarkers HDAC1, MDM2, MRPS10 and RPL7L1 in the ASF tissues were consistent with the results of the RNA-seq (Fig. 5D), confirming the accuracy of potential obese-specific biomarkers. In addition, significant differences in mRNA expression levels of HDAC1 (P = 0.0167), MRPS10 (P = 0.0015) and RPL7L1 (P = 0.0081) between the obese-type (Saba pigs) and lean-type (Large white pigs) pigs were observed, but no significant differences in those of MDM2 (P > 0.05). Detailed information was shown in Table S9.

Discussion

ASFT is an important energy storage organ and endocrine organ, the content of ASFD could affect the performance of these functions. However, the specific functional roles and regulatory mechanism of ASFD are still poorly understood. In this study, we systematically analyzed the gene expression profiles from differing ASFTs to investigate the transcriptome characteristics using bioinformatics methods. Finally, four obese-specific biomarkers, some key pathways, ASFT-related ceRNA network and transcriptional regulatory network were identified. It is crucial to identify candidate genes and elucidate the underlying molecular mechanisms that influence ASFD. On one hand, these findings can serve as potential biomarkers for early age selection in relation to meat quality. On the other hand, it is imperative in aiding our exploration of potential therapeutic targets for metabolic diseases associated with obesity. Among the identified key pathways of DEGs, three pathways related to lipid metabolism and adipocyte differentiation were found, including the p53 signaling pathway, MAPK signaling pathway, and fatty acid metabolism. A previous study showed that the p53 pathway was activated in mature obese adipocytes (Vergoni et al., 2016), and p53, the core gene in p53 signaling pathway, played a key role in regulating cellular metabolism (Berkers et al., 2013). Moreover, p53 activation in mice during high-fat diet (HFD) feeding could lead to HFD-induced obesity by regulating systemic metabolism (Liu et al., 2017). In our study, DEGs enriched in this pathway were basically significantly up-regulated. Such as MDM2, which targeted p53 for nuclear export and proteasomal degradation through attaching ubiquitin moieties, played a critical role in the maintenance of adipocyte homeostasis (Hallenborg et al., 2021). The MAPK signaling pathway consists of ERK, JNK and p38 signaling pathways. ERK-MAPK signaling pathway affects adipogenic differentiation (Poleti et al., 2018), and P38-MAPK was identified to be a positive regulator of intramuscular fat (IMF) deposition in pigs (Wu et al., 2017). The JNK-MAPK signaling pathway was associated with obesity in broilers muscle, and its inactivation could effectively resist obesity (Yan et al., 2013). The fatty acid metabolism signal pathway was involved in fatty acid biosynthesis and fatty acid beta oxidation, and played an important role in regulating fatty acid metabolism and growth traits in pigs (Yang et al., 2012). Several studies have shown that genes significantly up-regulated within the fatty acid metabolism signal pathway, such as ACAT1 (Huang et al., 2018), HADH (Kapoor, James & Hussain, 2009), ACADSB (Wang et al., 2015), were associated with energy metabolism. In our study, it was worth noting that the up-regulated genes (such as ACAT1 and HADH) of the fatty acid metabolism pathway were enriched in BPs such as fatty acid β-oxidation and negative regulation of lipid localization. This study further demonstrated that p53 signaling pathway, MAPK signaling pathway and fatty acid metabolism might play important roles in regulating ASFD in pigs.

Based on the GEO dataset, 10 hub genes (ABL1, HDAC1, CDC42, HDAC2, MRPS5, MRPS10, MDM2, JUP, RPL7L1 and UQCRFS1) were identified by the PPI network analysis with the algorithm of degree centrality. Meanwhile, KEGG pathway analysis indicated that these 10 hub genes were involved in immune response, fatness and metabolism. Published studies have shown that some genes in notch signaling pathway modulated the adipogenesis process. For example, Notch1 involved in the proliferation and differentiation of adipocyte progenitor cells in adipocyte progenitor cells (Bi et al., 2014; Chartoumpekis et al., 2015). Furthermore, Notch1 was identified to play a crucial role in the development and functions of MAs, beige adipocyte formation and energy metabolism (Bi & Kuang, 2015; Bi et al., 2016). Our study also identified that HDAC1 and HDAC2 in notch signaling pathway were differentially expressed between differing types of ASFTs, which indicated that notch signaling pathway might play a key role in ASFD.

Furthermore, the genes HDAC1, MDM2, MRPS10 and RPL7L1 were indicated to be potential robust obese-specific biomarkers among ten hub genes by the single gene expression analysis and ROC curves validation. Additionally, qRT-PCR results showed that HDAC1 was significantly up-regulated while MRPS10 and RPL7L1 were significantly down-regulated in verified obese-type pigs (Saba pigs), suggesting a solid difference in these potential biomarkers between obese and lean pigs. HDAC1 is involved in histone acetylation and deacetylation, catalyzed by multisubunit complexes, which plays a key role in the regulation of eukaryotic gene expression. Previous research has shown that high expression levels of HDAC1 were correlated with obesity and overweight in HFD-fed mice (Choi et al., 2017), while there was also a study showing the expression of HDAC1 in adipose tissues from obese women was lower in comparison with normal-weight individuals (Pour et al., 2020). And HDAC1, belongs to class I HDACs, is highly homologous and functionally redundant with HDAC2. HDAC2 plays an important role in transcriptional regulation, cell cycle progression and developmental events. It is an important paralog of HDAC1 (De Ruijter et al., 2003). A previous study also demonstrated that HDAC2 contributed to obesity (Nishimura et al., 2015). It was reported that HDAC2 mRNA expression in subcutaneous adipose tissue (SAT) was inversely correlated with waist circumference by comparing gene expression of HDAC2 in SAT between obese and non-obese women (Shanaki et al., 2020). However, some studies also reported that HDAC2, as the mediator of MKP-3 action in liver lipid metabolism, might be associated with reducing adiposity by repressing adipocyte differentiation in mice (Feng et al., 2014). HDAC1 also plays a crucial role in immune and inflammation regulation by promoting regulatory CD4 (+) T cells, CD8 (+) T cells (Boucheron et al., 2014; Tschismarov et al., 2014). In our study, significant upregulation of HDAC1 and HDAC2 was observed in the MAs of obese pigs compared to lean pigs. Overall, our results were consistent with previous studies indicating that HDAC1 and HDAC2 might play a critical role in fat deposition. MDM2 encodes a nuclear-localized E3 ubiquitin ligase, and MDM2 targeted p53 is associated with adipocyte homeostasis for nuclear export and proteasomal degradation through attaching ubiquitin moieties (Hallenborg et al., 2016a; Tyner et al., 2002). Some published studies showed that MDM2 played a pivotal role in the early steps of adipocyte differentiation (Hallenborg et al., 2012, 2016b). Recently, a study showed that the levels of MDM2 expression significantly increased in white adipose tissue (WAT) of diet-induced obese mice and genetically obese mice (Hallenborg et al., 2021). In addition, the previous study observed that MDM2 could mediated the degradation of p53 based on the rRNA transcription enhancement by IL-6, and that the degradation of p53 played a vital role in the process of cell transformation in inflamed tissues (Brighenti et al., 2014). MRPS10, related to peptide chain elongation and mitochondrial translation, has been reported to be associated with various diseases such as breast cancer and rheumatoid arthritis (Paramasivam et al., 2021). Published studies have reported that MRPS10 was potentially related with diseases such as Cardiovascular disease and obesity (Gopisetty & Thangarajan, 2016). Up to now, we have known little about the function of MRPS10 in fat deposition. RPL7L1 enables RNA binding activity, and is predicted to be involved in the maturation of LSU-rRNA from tricistronic rRNA transcript. Fewer studies were available on RPL7L1 (Thomas et al., 2014), and no published studies reported that the function of RPL7L1 was associated with fat deposition. Our results observed that low mRNA expression of RPL7L1 contributed to obesity in pigs, providing a novel insight into its role in fat deposition, and this needs to be further explored and confirmed.

Four genes, HDAC1, MDM2, MRPS10 and RPL7L1 were identified to be involved in fatty acids oxidation-related pathways such as propanoate metabolism and ribosome. In addition, several inflammation-related signaling pathways were identified such as butanoate metabolism, retinol metabolism, T cell receptor signaling pathway and oxidative phosphorylation. Retinol, as an indirect antioxidant, increases effective antioxidant response by affecting gene expression (Blaner, Shmarakov & Traber, 2021). A previous study has shown that increased plasma retinol might be associated with inflammatory dyslipidemia (Wang et al., 2021). T-cells, as vital effectors of cell-mediated immunity, can induce endothelial production of chemokines and cytokines (MCP1, IL-8 and IL-6) through T cell receptor activation or cytokines activation (Monaco et al., 2004; Tsai et al., 2018). T cell receptor-activated lymphocytes could cause monocytes to produce both the proinflammatory cytokine TNF and the anti-inflammatory cytokine IL-10. Inflammatory cytokines could regulate the capacity of TCR-signaling contributing to T-cell-intrinsic increases in antigen sensitivity and in vivo cytolytic capacity (Monaco et al., 2004). Metabolism process such as fatty acid oxidation and oxidative phosphorylation, is a critical immune regulator, could regulate immune cells and adaptive immune cells according to their activation/differentiation state (Stathopoulou, Nikoleri & Bertsias, 2019). The above results indicated that these genes played important role in regulating inflammation and immunity.

In addition, some key lncRNAs involved in fat deposition were identified in the ceRNA network, such as NEAT1 and XIST. LncRNA NEAT1 plays an essential role in regulating cellular function during development and metabolic processes (Clemson et al., 2009). It has been reported that NEAT1 is involved in alternative splicing of PPARγ mRNA, thereby working in the timing of alternative splicing of primary transcripts to regulate adipogenesis (Cooper et al., 2014). LncRNA XIST is associated with the cell differentiation, proliferation, X-chromosome inactivation, and immunity (Payer & Lee, 2008; Yildirim et al., 2013). A recent study showed that XIST played a critical role in brown preadipocytes differentiation and metabolic regulation (Wu et al., 2022). These results are consistent with previous findings indicating that lncRNAs NEAT1 and XIST may be novel potential regulatory factors in ASFD.

Although four obese-specific biomarkers were identified and some genes have been validated, some limitations must be noted in the current study. First, the results were obtained by a bioinformatics method with small sample size, the expression level of biomarkers must be validated using larger sample sizes and more pig breeds via more accurate methods. Second, the specific functions of biomarkers and targeted lncRNAs in ceRNA need to be revealed in ASFD by overexpression or knockdown methods.

Conclusions

Taken together, our study systematically analyzed gene expression data related to fat deposition using a comprehensive bioinformatics method, and identified ten hub genes and several pathways associated with ASFD in pigs, especially the hub genes, HDAC1, MDM2, MRPS10 and RPL7L1 might be as potential obese-specific biomarkers. These findings provided novel insights into the molecular mechanism involved in ASFD.

Supplemental Information

Supplemental Information 1 Supplemental tables.

Supplemental Information 2 Author Checklist.

Supplemental Information 3 MIQE checklist.

Supplemental Information 4 Raw data.

Supplemental Information 5 qPCR raw data.

Additional Information and Declarations

Competing Interests

Author Contributions

Animal Ethics

Data Availability

The authors declare that they have no competing interests.

Yongli Yang conceived and designed the experiments, performed the experiments, analyzed the data, prepared figures and/or tables, and approved the final draft.

Xiaoyi Wang conceived and designed the experiments, analyzed the data, prepared figures and/or tables, and approved the final draft.

Mingli Li conceived and designed the experiments, analyzed the data, prepared figures and/or tables, and approved the final draft.

Shuyan Wang performed the experiments, analyzed the data, prepared figures and/or tables, and approved the final draft.

Huiyu Wang performed the experiments, analyzed the data, prepared figures and/or tables, and approved the final draft.

Qiang Chen conceived and designed the experiments, analyzed the data, prepared figures and/or tables, authored or reviewed drafts of the article, and approved the final draft.

Shaoxiong Lu conceived and designed the experiments, authored or reviewed drafts of the article, and approved the final draft.

The following information was supplied relating to ethical approvals (i.e., approving body and any reference numbers):

The experimental protocol was approved by the Animal Ethics Committee of Yunnan Agricultural University (approval ID: 202310003).

The following information was supplied regarding data availability:

The raw measurements are available in the Supplemental Files.

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
