# Peer review of "Identification of potential obese-specific biomarkers and pathways associated with abdominal subcutaneous fat deposition in pig using a comprehensive bioinformatics strategy"

_PeerJ, doi:10.7717/peerj.17486_

## Round 0.1 · original submission · Major Revisions

Please address the concerns of all reviewers and amend the manuscript accordingly.

**Language Note:** PeerJ staff have identified that the English language needs to be improved. When you prepare your next revision, please either (i) have a colleague who is proficient in English and familiar with the subject matter review your manuscript, or (ii) contact a professional editing service to review your manuscript. PeerJ can provide language editing services - you can contact us at [email protected] for pricing (be sure to provide your manuscript number and title). – PeerJ Staff

Reviewer 1 ·

Basic reporting

The manuscript is well-structured, adhering to the required standards with clear and professional English language usage throughout the text.
The introduction provides a good context for the study, explaining the significance of abdominal subcutaneous fat deposition (ASFD) in pigs both for the meat industry and its relevance to human health.
Literature is well-referenced, offering a solid background and demonstrating a gap in the research that this study aims to fill.
Figures and tables appear to be appropriate for the content, with the raw data being made available in accordance with the journal's policy, which is commendable.

Experimental design

The research fits within the scope of the journal, focusing on identifying obese-specific biomarkers and key pathways related to ASFD in pigs using a bioinformatics approach.
The research question is well-defined and meaningful, with the methods described in sufficient detail for replication.
The study design seems rigorous, using datasets from GEO, and employing various bioinformatics tools for the analysis, thus meeting high technical and ethical standards.

Validity of the findings

The research findings are robust, supported by sound statistical analysis. The study identifies differentially expressed genes (DEGs) and constructs a protein-protein interaction (PPI) network, lending significant weight to the conclusions. The link between the findings and the original research question is well-maintained throughout, with the identified biomarkers undergoing rigorous validation processes.

Additional comments

To further strengthen the study, it would be beneficial to consider a larger and more varied sample size across different pig breeds. A more detailed rationale for the selection of specific bioinformatics tools and parameters would also enhance the manuscript. Lastly, expanding on the practical implications of these biomarkers for the meat industry and human health would be a valuable addition to the discussion.

Reviewer 2 ·

Basic reporting

The GEO showed up in the abstract. It needs to be spelled out the first time it shows up in your manuscript.

In the section ‘construction of ceRNA network’, the mRNA-miRNA relationship pairs were extracted based on at least two of the databases, miRanda (John et al., 2004), TargetScan (Agarwal et al., 2015), RNA22 (Loher & Rigoutsos, 2012), and miRmap (Vejnar & Zdobnov 2012). Why did you list four databases but state “at least two of the databases”?

In Figure 6, the labels were overlapped with the lines.

Experimental design

Considering your study sample size, it may not be appropriate to draw ROC curve and generate conclusions from statistical analysis results based on large sample assumptions.

How was the mRNA-miRNA-lncRNA ceRNA network validated by the six-pig (3 Saba, 3 Large white pigs) study, or were they unrelated?

Validity of the findings

How many pigs were studied in the GEO database? The results and conclusions from 6 pigs may not sound solid enough unless it is proven to be a generally acceptable practice.

It is unclear which biomarkers were identified through the Gene Expression Omnibus (GEO) database, other databases, or the six pigs study. This should be written in the introduction section and emphasized in each biomarker identification section.

Considering your study sample size, it may not be appropriate to draw the ROC curve and generate conclusions from statistical analysis results based on large sample assumptions.

Reviewer 3 ·

Basic reporting

While this paper is clear, the discussion regarding its integration into future work is limited. No novel methods have been developed. Raw data is shared, and the link for the raw data is verified. However, the figures are not organized clearly. I recommend that you discuss the usage of your method more extensively. Also please report the percentage of variation explained by each functional component. Additionally, consider plotting the eigenfunctions of each component to visually depict the variation.

Experimental design

No comment.

Validity of the findings

All data is provided, but the tables are not summarized in a clear way. Consider adding comprehensive titles to the tables that can explain their content more clearly.

Additional comments

Please check your grammar, there are some obvious grammatical errors. For example, in line 127, it should be 'was considered as a potential biomarker'.

---

## Round 0.2 · accepted · Accept

All issues pointed out by the reviewers were adequately addressed and the revised manuscript is acceptable now.

Reviewer 3 ·

Basic reporting

The author(s) have addressed my previous comments and no further comments.

Experimental design

The author(s) have addressed my previous comments and no further comments.

Validity of the findings

The author(s) have addressed my previous comments and no further comments.